# Active Matrix Metalloproteinase-8 Point-of-Care (PoC)/Chairside Mouthrinse Test vs. Bleeding on Probing in Diagnosing Subclinical Periodontitis in Adolescents

**DOI:** 10.3390/diagnostics9010034

**Published:** 2019-03-23

**Authors:** Ismo T. Räisänen, Timo Sorsa, Gerrit-Jan van der Schoor, Taina Tervahartiala, Peter van der Schoor, Dirk-Rolf Gieselmann, Anna Maria Heikkinen

**Affiliations:** 1Department of Oral and Maxillofacial Diseases, Head and Neck Center, University of Helsinki and Helsinki University Hospital, P.O. Box 63 (Haartmaninkatu 8), 00014 Helsinki, Finland; timo.sorsa@helsinki.fi (T.S.); taina.tervahartiala@helsinki.fi (T.T.); anna.m.heikkinen@helsinki.fi (A.M.H.); 2Division of Periodontology, Department of Dental Medicine, Karolinska Institute, SE-171 77 Stockholm, Sweden; vdschoor.putten@gmail.com (G.-J.v.d.S.); peter@vanderschoor.org (P.v.d.S.); 3Institute for Molecular Diagnostics (IMOD), Bonner Str. 84, 42697 Solingen, Germany; gieselmann@matrix-lab.de

**Keywords:** adolescent, periodontitis, matrix metalloproteinases, point-of-care testing, gingival bleeding on probing

## Abstract

This cross-sectional study compares the effectiveness of an active MMP-8 (aMMP-8) point-of-care (PoC)/chairside mouthrinse test to the conventional bleeding on probing (BOP) (cutoff 20%) test in detecting subclinical periodontitis/pre-periodontitis in Finnish adolescents. The study was carried out at the Kotka Health Center, Finland. A total of 47 adolescents (30 boys/17 girls) aged 15–17 were first tested with the aMMP-8 PoC test, followed by a full-mouth evaluation of clinical parameters of oral health including periodontal, oral mucosal, and caries assessment. A periodontist performed these clinical examinations. The aMMP-8 PoC test result had much stronger association with subclinical periodontitis than the BOP 20% test (2.8–5.3 times stronger in terms of odds ratio). The aMMP-8 PoC test had ≥2 times higher sensitivity than the BOP 20% test with, generally, the same specificity. Further, the aMMP-8 PoC test had generally better accuracy and lower false negative percentages. The aMMP-8 PoC test seemed to be more effective than the conventional BOP test in detecting subclinical periodontitis/pre-periodontitis in adolescents reducing the risk of their undertreatment. However, the sample size may be a limiting factor, and more studies are needed to confirm our results for both adolescents and adults.

## 1. Introduction

Activated matrix metalloproteinase-8 (MMP-8) is a major destructive collagenase in periodontitis [1] but, also, a potential biomarker for initial periodontitis/subclinical periodontitis/pre-periodontitis as well as periodontitis [1,2,3,4,5]. According to several studies [1,2,6,7,8,9,10,11], the progression of periodontitis is reflected as an excessive elevation and activation of MMP-8 in oral fluids. This stage can be measured using a point-of-care lateral flow collagenase-2 immunotest [1,12,13,14,15,16,17,18,19,20,21,22,23]. Currently, this active MMP-8 (aMMP-8) point-of-care (PoC)/chairside mouthrinse test exists and is available as a visual and a quantitative point-of-care aMMP-8 test [13].

Bleeding on probing (BOP) has been widely used as a sign of gingival inflammation and to determine the periodontal health of a patient [24]. This has been done with the periodontal probe when assessing the periodontal status. However, there are some problems related to measuring the bleeding on probing, most importantly, the risk of experiencing bacteremia as well as the variation in the operator technique. Practically, probing force is not the same for all clinicians which, unfortunately, makes bleeding on probing a somewhat subjective measure. Moreover, periodontal probing contributes to the risk for developing infective endocarditis and, thus, antibiotic prophylaxis is recommended for medically compromised patients [25,26]. Antibiotic resistance is a major problem and, thus, new approaches are needed in this regard as well [27].

Thus, there is a real incentive to develop new and better methods to determine periodontal health. The aMMP-8 PoC test has been validated in different ethnic populations as a reliable tool to differentiate periodontal health and disease [12,13,14,15,16,17,18,19,20,21,22,23]. It is inexpensive, quick and easy to use, and, most importantly, a standardized, non-invasive, and safe point-of-care test, which makes it a promising candidate as a chairside test for periodontal diseases. Previously, the aMMP-8 PoC test has shown some promising results in the detection of oral inflammatory burden in adolescents with early signs of initial periodontitis/subclinical periodontitis [15,16]. In this study, our aim was to compare the effectiveness of an aMMP-8 PoC mouthrinse test to the conventional BOP test (cutoff point 20% for positives) in the diagnostics of initial periodontitis/subclinical periodontitis in Finnish adolescents.

## 2. Materials and Methods

This cross-sectional study was carried out in Kotka Health Center, Kotka, Eastern Finland in 2014 and 2015. It was approved by the Ethical Committee of Helsinki University Central Hospital, Helsinki, Finland (Diary number 260/13/03/00/13, 17.12.2013). The study was conducted according to the principles of the Declaration of Helsinki (2008) [28]. A total of 47 adolescents (30 males and 17 females, aged between 15 and 17 years) provided written informed consent and participated in this study. One participant was excluded after a discrepant gender check. For reasons that remained unknown, the rest of the birth cohort (73 adolescents) refused to participate.

As described in Heikkinen et al. [15,16,29], first, the aMMP-8 chairside lateral flow mouthrinse immunoassay test (PerioSafe, Dentognostics GmbH, Jena, Germany) was performed by a periodontist (AMH) in accordance with the manufacturer’s instructions. The aMMP-8 mouthrinse test results were read within 5 minutes by eye, based on the color change resulting from immunoreactions: one single blue line for a negative result (no risk) and two blue lines for a positive result (increased risk). Even a thin second line indicates increased risk for periodontitis and peri-implantitis. Secondly, clinical parameters were collected from the patients by an AMH, e.g., BOP ≥ 20% (cutoff point 20% for positives as described in Heikkinen et al. [30,31]) was recorded as well as probing depth: (PD)1 (one or more ≥4 mm periodontal pocket), PD2 (two or more ≥4 mm periodontal pockets), PD3 (three or more ≥4 mm periodontal pockets), PD4 (four or more ≥4 mm periodontal pockets). Other parameters, like caries (D) and visible plaque index (VPI%), were also recorded. The AMH was a priori unaware of the aMMP-8 chairside mouthrinse test results of the subjects when performing the clinical examinations. Finally, an oral radiologist analyzed right and left bitewing radiographs and evaluated the condition of alveolar bone in the interproximal area of the premolars and molars [29].

### Statistical Analysis

We used 2 × 2 contingency tables (cross tabulations) to test the performance of two tests—the aMMP-8 PoC test and the BOP test (cutoff point 20% for positives)—in predicting the presence of initial periodontitis or subclinical periodontitis (4 cases, PD1–PD4; see the frequencies in Table 1) among adolescents. The healthy controls were defined as adolescents that have no ≥4 mm periodontal pockets (see their frequencies in Table 1). We calculated the diagnostic odds ratio (diagnostic OR), sensitivity (true positive rate), specificity (true negative rate), accuracy (the percentage of correctly classified instances), false discovery rate/the percentage of false positives (FP(%)), and false omission rate/the percentage of false negatives (FN(%)). We also approximated the receiver operating characteristic (ROC) curves and the area under the ROC curve (AUC) for both tests in each case (trapezoid method, three reference points). Analysis was performed with Microsoft^®^ Excel^®^ for Mac 2011.

## 3. Results

The prevalence of the four different initial periodontitis or subclinical periodontitis cases (the number of sites with probing depth (PD) ≥4 mm; PD1, PD2, PD3 and PD4) was quite high: 61.7% for PD1, 46.8% for PD2, 36.2% for PD3, and 34.0% for PD4. Diagnostic odds ratios (OR) were higher for the aMMP-8 PoC test in all four cases compared to the conventional BOP test (cutoff point 20% positive) (Table 1): for PD1, OR = 34.6 and OR = 12.3; for PD2, OR = 63.1 and OR = 17.9; for PD3, OR = 111.0 and OR = 20.9; and for PD4, OR = 102.8 and OR = 22.9 (the aMMP-8 PoC test and BOP (20%) test, respectively). Similarly, the aMMP-8 PoC test had higher sensitivity than the BOP (20%) test in all four cases (Table 1). For PD1, the PoC test sensitivity = 48.3% and the BOP (20%) test sensitivity = 24.1%; for PD2, the PoC test sensitivity = 63.6% and the BOP (20%) test sensitivity = 31.8%; for PD3, the PoC test sensitivity = 76.5% and the BOP (20%) test sensitivity = 35.3%; and for PD4, the PoC test sensitivity = 75.0% and the BOP (20%) test sensitivity = 37.5%. However, test specificities were the same (100.0%) for both tests (Table 1).

The ROC curve and the AUC analysis for the aMMP-8 PoC test and the conventional BOP (20%) test is presented in Figure 1. The aMMP-8 PoC test had higher approximated AUC values in all PD1–PD4 cases compared to the BOP (20%) test: for PD1, AUC = 0.741 and AUC = 0.621 (PoC and BOP (20%), respectively); for PD2, AUC = 0.818 and AUC = 0.659 (PoC and BOP (20%), respectively); for PD3, AUC = 0.882 and AUC = 0.676 (PoC and BOP (20%), respectively); for PD4, AUC = 0.875 and AUC = 0.688 (POC and BOP (20%), respectively). The approximated AUC values are the lower-bound estimates for both tests.

Finally, the aMMP-8 PoC test had higher test accuracy than the conventional BOP (20%) test in all PD1–PD4 cases (Table 1): for PD1, the PoC test accuracy = 68.1% and the BOP (20%) test accuracy = 53.2%; for PD2, the PoC test accuracy = 80.0% and the BOP (20%) test accuracy = 62.5%; for PD3, the PoC test accuracy = 88.6% and the BOP (20%) test accuracy = 68.6%; and for PD4, the PoC test accuracy = 88.2% and the BOP (20%) test accuracy = 70.6%. The false omission rate (FN(%)) was lower for the aMMP-8 PoC test than the conventional BOP (20%) test in all PD1–PD4 cases (Table 1): for PD1, PoC FN% = 45.5% and BOP (20%) FN% = 55.0%; for PD2, PoC FN% = 30.8% and BOP (20%) FN% = 45.5%; for PD3, PoC FN% = 18.2% and (20%) BOP FN% = 37.9%; and for PD4, PoC FN% = 18.2% and BOP (20%) FN% = 35.7%. The false discovery rates (FP(%)) were equal for both tests (0.0%) (Table 1).

## 4. Discussion

The main finding of this study was that the aMMP-8 PoC test had a much better ability to correctly classify adolescents with initial periodontitis or subclinical periodontitis/pre-periodontitis as diseased compared to the conventional BOP (cutoff point 20% for positives) test. The aMMP-8 PoC test had 2.8–5.3 times stronger association (in terms of diagnostic odds ratio) with initial periodontitis or subclinical periodontitis compared to the BOP test in all our cases (one or more ≥4 mm deep periodontal pockets, PD1; two or more ≥4 mm deep periodontal pockets, PD2; three or more ≥4 mm deep periodontal pockets, PD3; and four or more ≥4 mm deep periodontal pockets, PD4). By adolescents’ initial periodontitis, we refer to the subclinical periodontitis, where the clear clinical and X-ray manifestations of deterioration of periodontal health are low or lacking. This could be seen as an early stage of the disease, basically corresponding to the stage I or even a subclinical state (before stage I) of periodontitis under the new classification of periodontal diseases [34].

Our current findings suggest that aMMP-8 PoC/chairside test alerts to this pathological state more efficiently than BOP assessments. The risk of patient overtreatment was low for both tests due to their high test specificity and low false discovery rate (i.e., false prediction of disease), but the aMMP-8 PoC test had at least two times higher test sensitivity compared to the BOP test in all four cases. Due to this better test sensitivity, the risk of patient undertreatment seems to be much lower when using the aMMP-8 PoC test compared to the BOP test. It should be noted that diagnostic odds ratio, sensitivity, and specificity results are not dependent on disease prevalence, as is the case with test accuracy, false omission rate, and false discovery rate. Similarly, the calculated ROC curves and AUCs suggest that the aMMP-8 PoC test is more accurate for classifying adolescents with and without initial or subclinical periodontitis than the BOP test.

There are some limitations related to the use of the aMMP-8 PoC test, namely, patients with mixed dentition, i.e., patients younger than 15 years old, an active phase in orthodontic treatment, pericoronitis, or a mouth ulcer. They are potential sources of error for the aMMP-8 PoC test in the identification of patients for periodontal diseases and may cause false positive test results. In this regard, Schmidt et al. [35] demonstrated a statistically significant relationship between the aMMP-8 PoC test positive result and PD ≥ 4mm (the community periodontal index of treatment needs, CPITN > 2), as an early indicator for periodontal disease among adolescents between 10 and 18 years of age. In their study, the sensitivity and the specificity of aMMP-8 PoC test were 61% and 52%, respectively, but the large amount of young patients with mixed dentition (aged between 10 and 15 years) (*n* = 372) and also the large amount of adolescents with orthodontic appliances (*n* = 357) eventually affected the recorded sensitivity and specificity, in comparison to our findings using only 15–17-year-old adolescents without mixed dentition and who were not under orthodontic treatment. Notably, orthodontic treatment is known to increase aMMP-8 in oral fluids without pocket formation [36,37].

In this study, we also found that the prevalence of deep pockets was quite high in our sample of adolescents, suggesting that potentially many adolescents are at risk for subclinical periodontitis. Previously, Heikkinen et al. [29] also reported that a total of 72% (*n* = 33) of all adolescents in their study had suspect radiographic signs of periodontitis and 37% (*n* = 17) had horizontal bone loss (distance between alveolar crest and cementoenamel junction of more than 2 mm). Thus, there is a real need for new tools like the aMMP-8 PoC test, which may help to target the periodontal prevention more efficiently. BOP seems to be a more erroneous method of diagnosis, leading to patient undertreatment in comparison with the aMMP-8 PoC test. Further, measuring BOP always causes bacteremia, whereas the aMMP-8 PoC test is a non-invasive diagnostic method that never requires antibiotic prophylaxis. Therefore, using the aMMP-8 PoC test may also decrease the use of antibiotics and reduce the spread of antibiotic resistance.

The antibody utilized by the aMMP-8 PoC test identifies specifically activated and fragmented forms of MMP-8, which are characteristics of, and specific to, active periodontitis and peri-implantitis sites/lesions, differentiating from gingivitis and healthy tissues [38,39,40]. Neutrophilic MMP-8 is secreted, or degranulated, in the inactive pro-form having both pro-domain and the C-terminal hemopexin domain 1, and the cleavage of pro-domain triggers pro-form activation and reduces it to its active form; as a result, the C-terminal domain is lacking in the 20–27 kDa MMP-8 fragment after MMP-8 activation [9,38,39]. Both Uitto et al. [40] and Gangbar et al. [41] have independently demonstrated that MMP-8 in oral saliva and in mouthrinse are both derived from polymorphonuclear neutrophils that enter sulcular/gingival crevicular fluid through gingival sulcus and not from other sources, e.g., salivary glands and tonsillar or oral mucosa; edentulous subjects exhibit neither active nor latent MMP-8 in their saliva/mouthrinse samples. In addition to the aMMP-8 lateral flow PoC immunotest, the concentration of aMMP-8 can also be measured using standard laboratory methods by enzyme-linked immunoassay (ELISA) or immunofluorometric assay (IFMA). In this regard, when the same MMP-8 antibodies are used, there is a strong agreement between ELISA and IFMA, but also between the aMMP-8 PoC test and ELISA, and the aMMP-8 PoC test and IFMA [17,23,42]. All this supports the usefulness of the aMMP-8 PoC test in the diagnosis and assessment of periodontal disease activity.

We believe that the aMMP-8 PoC test is a promising tool that can help oral health care professionals to more reliably identify adolescents who are at risk for subclinical periodontitis/pre-periodontitis. Currently, the PoC test exists and is also available as a quantitative point-of-care aMMP-8 test [13]. The aMMP-8 PoC test also has the potential to enhance the periodontal risk assessment of patients. As has been suggested by Heikkinen et al. [2], the aMMP-8 PoC test result could be used together with other patient-related risk factor information, e.g., gender, use of tobacco products, as well as oral health habits, such as toothbrushing, to create an even more accurate periodontal disease risk profile for patients. Moreover, Räisänen et al. [22] recently reported a strong association between patient’s aMMP-8 PoC test results and his/her periodontal treatment needs (CPITN) among adolescents and adults. In their study, the aMMP-8 PoC test did not cause any patient overtreatment among adolescents and adults [22], which is in agreement with our findings. Overall, our results are promising, even if the sample size (*n* = 47) is taken into account. In the long run, the aMMP-8 PoC test could be a cost-efficient and easy method to be used also in subclinical periodontitis/pre-periodontitis diagnostics, but more research is still needed to confirm this.

Finally, some alternative methods using other potential biomarkers in oral fluids have been under development for diagnostics of periodontitis and peri-implantitis [43,44,45]. These biomarkers include, but are not limited to, neutrophil elastase, interleukin-1β, myeloperoxidase, proteases of *Porphyromonas gingivalis* and *Treponema denticola* (gingipains and dentilisin), and calprotectin [43,44,45]. Using them in combination may increase the accuracy of periodontitis diagnosis compared to a single biomarker. Unfortunately, these methods are still at the experimental stage. On the other hand, visual and quantitative aMMP-8 PoC/chairside assay, which function very well on their own, are commercially available for dental and medical professionals linking these two disciplines.

## 5. Conclusions

In summary, our study demonstrates the effectiveness of the non-invasive aMMP-8 PoC/chairside mouthrinse test in diagnosing initial periodontitis or subclinical periodontitis in adolescents compared to BOP. The aMMP-8 PoC test had a much stronger association with subclinical periodontitis than BOP, and even though they both had a very high ability to correctly identify healthy adolescents, the aMMP-8 PoC test had at least two times higher sensitivity for identifying adolescents with subclinical periodontitis. Thus, the aMMP-8 PoC test seemed to be far more accurate than BOP in detecting subclinical periodontitis in adolescents, reducing the risk of undertreatment.

## Figures and Tables

**Figure 1 diagnostics-09-00034-f001:**
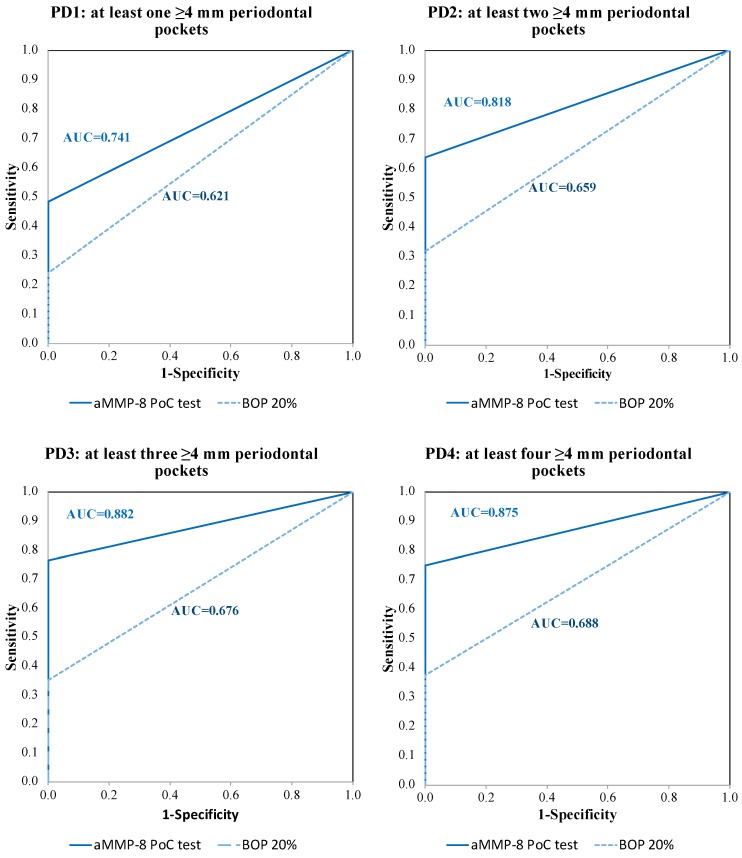
Receiver operating characteristic (ROC) curves and the area under the ROC curve (AUC) values for the aMMP-8 PoC test and the conventional BOP test (cutoff point 20% for positives) representing the ability of both tests to classify adolescents with and without initial periodontitis or subclinical periodontitis (PD1–PD4).

**Table 1 diagnostics-09-00034-t001:** Frequencies of the positive and negative test results of active MMP-8 (aMMP-8) point-of-care (PoC) and bleeding on probing (BOP) (cutoff point 20% for positives) tests among Finnish adolescents according to the four initial periodontitis or subclinical periodontitis cases (PD1–PD4). Diagnostic OR, sensitivity, specificity, test accuracy, and the percentage of false negatives and false positives were calculated for both tests.

	Periodontal Status		Measures of the Effectiveness of a Diagnostic Test					
Test	PD1	Healthy	OR	Se	Sp	Acc	FN	FP
aMMP-8 PoC+	14	0	34.6	48.3%	100.0%	68.1%	45.5%	0.0%
aMMP-8 PoC−	15	18						
BOP+	7	0	12.3	24.1%	100.0%	53.2%	55.0%	0.0%
BOP−	22	18						
Test	PD2	Healthy	OR	Se	Sp	Acc	FN	FP
aMMP-8 PoC+	14	0	63.1	63.6%	100.0%	80.0%	30.8%	0.0%
aMMP-8 PoC−	8	18						
BOP+	7	0	17.9	31.8%	100.0%	62.5%	45.5%	0.0%
BOP−	15	18						
Test	PD3	Healthy	OR	Se	Sp	Acc	FN	FP
aMMP-8 PoC+	13	0	111.0	76.5%	100.0%	88.6%	18.2%	0.0%
aMMP-8 PoC−	4	18						
BOP+	6	0	20.9	35.3%	100.0%	68.6%	37.9%	0.0%
BOP−	11	18						
Test	PD4	Healthy	OR	Se	Sp	Acc	FN	FP
aMMP-8 PoC+	12	0	102.8	75.0%	100.0%	88.2%	18.2%	0.0%
aMMP-8 PoC−	4	18						
BOP+	6	0	22.9	37.5%	100.0%	70.6%	35.7%	0.0%
BOP−	10	18						

aMMP-8 PoC+ = PerioSafe^®^ test positive; aMMP-8 PoC− = PerioSafe^®^ test negative; BOP+ = bleeding on probing over 20%; BOP− = bleeding on probing less than or equal 20%. PD1 = the number of adolescents that have at least one ≥4 mm periodontal pocket; PD2 = the number of adolescents that have at least two ≥4 mm periodontal pockets; PD3 = the number of adolescents that have at least three ≥4 mm periodontal pockets; PD4 = the number of adolescents that have at least four ≥4 mm periodontal pockets; Healthy = the number of adolescents that do not have any ≥4 mm periodontal pockets. OR = diagnostic odds ratio, calculated using Haldane–Anscombe correction (0.5 added to each cell) [32,33]. Se = sensitivity; Sp = specificity; Acc = accuracy; FN = the percentage of false negatives; FP = the percentage of false positives.

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
