# Peer review of "Active Matrix Metalloproteinase-8 Point-of-Care (PoC)/Chairside Mouthrinse Test vs. Bleeding on Probing in Diagnosing Subclinical Periodontitis in Adolescents"

_diagnostics, 2019, doi:10.3390/diagnostics9010034_

Round 1
Reviewer 1 Report
Authors shows that the aMMP-8 PoC test has a better ability to detect subclinical periodontitis in adolescents compared to the conventional BOP. The results demonstrated that aMMP-8 PoC test had larger AUCs with initial or subclinical periodontitis than the BOP test in all probing depth groups (PD1-PD4).
I agree the authors’ conclusion, but there are some points that need to be addressed.
1. With regard to aMMP-8 PoC test kit, please explain the principles behind aMMP-8 PoC test and accuracy compared to ELISA. Active form of MMP-8 lacks the C-terminal hemopexin domain 1. Does this antibody conjugated to the kit detect active form specifically or detect C-terminal domain?
2. In Table 1, what does “N” group represent? It's hard to understand the categories (PD1, #PD1, PD2, #PD2...) and what authors want to show in this table. Legend is not sufficient.
3. Authors need to include healthy control group and compare the AUC to that of PD group.
4. aMMP can be secreted from salivary gland and oral mucosa, meaning aMMP in saliva is not only an indicator of periodontitis but also represents the physical condition and background diseases. I would suggest authors add discussion about this possibility.
Author Response
Dear Reviewer 1,
Thank you for your comments!
Please find our responses to your comments below. They have been described in the revised manuscript, as well.
Yours faithfully,
Ismo Räisänen
Point 1: “With regard to aMMP-8 PoC test kit, please explain the principles behind aMMP-8 PoC test and accuracy compared to ELISA. Active form of MMP-8 lacks the C-terminal hemopexin domain 1. Does this antibody conjugated to the kit detect active form specifically or detect C-terminal domain?”
Response 1: The test (PerioSafe®) is a point-of-care / chairside lateral flow collagenase-2 (aMMP-8) immunotest and is available as a visual and a quantitative test. This study used the visual test. Both tests (the visual and the quantitative) analyze patient’s mouthrinse and, thus, are a non-invasive diagnostic method causing no discomfort to patient. Mouthrinse was collected in three phases according to manufacturer’s instructions): first a 30 second prerinse with tap water, followed by a 60 second wait, and then rinsing with the test liquid (5 mL purified water) for 30 seconds and pouring it into a little collection cup. The test administrator collects 3 mL of the rinse into a syringe and places maximum of four drops in the lateral flow immunoassay system. The result can be read in 5-7 minutes as a color change resulting from immunoreactions: one single blue line for a negative result (no risk) and two blue lines for positive result (increased risk). Even a thin second line indicates increased risk for periodontitis and peri-implantitis. In the quantitative test, the test analyzes the concentration of aMMP-8 in patient’s mouthrinse. The visual and the quantitative tests have the same cut-off point for positives 20ng/mL. [1,2]
In addition to the aMMP-8 lateral flow PoC immunotest, the concentration of aMMP-8 can be measured using standard laboratory methods by enzyme-linked immunoassay (ELISA) or immunofluorometric assay (IFMA). In this regard, when the same MMP-8 antibodies are used, there is a strong agreement between ELISA and IFMA, but also between the aMMP-8 PoC test and ELISA, and the aMMP-8 PoC test and IFMA. [1,3,4]
The antibody utilized by the aMMP-8 PoC test identifies specifically activated and fragmented forms of MMP-8; this activation and fragmentation is the characteristics of and specific to disease active periodontitis lesions / sites differing from gingivitis and healthy ones. Neutrophilic MMP-8 is secreted, or degranulated, as in the inactive pro-form having both pro-domain and the C-terminal hemopexin domain 1, and cleavage of pro-domain triggers pro-form activation and reduces it to its active form; eventually C-terminal domain is lacking in the 20-27 kDa MMP-8 fragment after MMP-8 activation. [5,6,7]
This (with all references) has been amended to manuscript page 2 lines 77-82 and page 6 lines 187-202.
[1] Alassiri S, Parnanen P, Rathnayake N, Johannsen G, Heikkinen AM, Lazzara R, van der Schoor P, van der Schoor JG, Tervahartiala T, Gieselmann D, Sorsa T. The Ability of Quantitative, Specific, and Sensitive Point-of-Care/Chair-Side Oral Fluid Immunotests for aMMP-8 to Detect Periodontal and Peri-Implant Diseases. Dis Markers. 2018 Aug 5;2018:1306396. doi: 10.1155/2018/1306396
[2] Al-Majid A, Alassiri S, Rathnayake N, Tervahartiala T, Gieselmann DR, Sorsa T. Matrix Metalloproteinase-8 as an Inflammatory and Prevention Biomarker in Periodontal and Peri-Implant Diseases. Int J Dent. 2018 Sep 16;2018:7891323. doi: 10.1155/2018/7891323
[3] Nieminen MT, Vesterinen P, Tervahartiala T, Kormi I, Sinisalo J, Pussinen PJ, Sorsa T. Practical implications of novel serum ELISA-assay for matrix metalloproteinase-8 in acute cardiac diagnostics. Acute Card Care. 2015;17(3):46-7. doi: 10.3109/17482941.2015.1115077
[4] Lorenz, K.; Keller, T.; Noack, B.; Freitag, A.; Netuschil, L.; Hoffmann. T. Evaluation of a Novel Point-of-Care Test for Active Matrix Metalloproteinase-8: Agreement between Qualitative and Quantitative Measurements and Relation to Periodontal Inflammation. J Periodontal Res 2017, 52, 277-284. https://doi.org/10.1111/jre.12392
[5] Sorsa T, Hernández M, Leppilahti J, Munjal S, Netuschil L, Mäntylä P. Detection of gingival crevicular fluid MMP-8 levels with different laboratory and chair-side methods. Oral Dis. 2010 Jan;16(1):39-45. doi: 10.1111/j.1601-0825.2009.01603.x
[6] Hanemaaijer R, Sorsa T, Konttinen YT, Ding Y, Sutinen M, Visser H, van Hinsbergh VW, Helaakoski T, Kainulainen T, Rönkä H, Tschesche H, Salo T. Matrix metalloproteinase-8 is expressed in rheumatoid synovial fibroblasts and endothelial cells. Regulation by tumor necrosis factor-alpha and doxycycline. J Biol Chem. 1997 Dec 12;272(50):31504-9. doi: 10.1074/jbc.272.50.31504
[7] Gürsoy UK, Könönen E, Tervahartiala T, Gürsoy M, Pitkänen J, Torvi P, Suominen AL, Pussinen P, Sorsa T. Molecular forms and fragments of salivary MMP-8 in relation to periodontitis. J Clin Periodontol. 2018 Dec;45(12):1421-1428. doi: 10.1111/jcpe.13024
Point 2: “In Table 1, what does “N” group represent? It's hard to understand the categories (PD1, #PD1, PD2, #PD2...) and what authors want to show in this table. Legend is not sufficient.”
Response 2: We have now put together the information of tables 1 and 2 to make the information overall easier to understand. See table 1 on page 3. It is supposed to give information on how the two tests classify patients with different number of ≥4 mm periodontal pockets i.e. signs of initial periodontitis / subclinical periodontitis compared to healthy controls.
Point 3: “Authors need to include healthy control group and compare the AUC to that of PD group.”
Response 3: We have now included healthy control group (adolescents without any ≥4 mm periodontal pockets) and updated our results section and information in the figure 1 and table 1 accordingly. See pages 3-5 lines 103-140.
Point 4: “aMMP can be secreted from salivary gland and oral mucosa, meaning aMMP in saliva is not only an indicator of periodontitis but also represents the physical condition and background diseases. I would suggest authors add discussion about this possibility.”
Response 4: Both Uitto et al. (1990) and Gangbar et al. (1990) have independently demonstrated that MMP-8 in oral saliva and in mouthrinse are both derived from PMNs that enter sulcular / gingival crevicular fluid through gingival sulcus and not from other sources e.g. salivary glands, tonsillar or oral mucosa; edentulous subjects exhibit neither active nor latent MMP-8 in their saliva / mouthrinse samples. This supports the usefulness of the aMMP-8 PoC test especially in the diagnosis and assessment of periodontal disease activity. However, in addition to oral health care professionals, medical professionals should be aware of the periodontal status of their patients as well. Periodontitis increases the risk of several general diseases like diabetes, stroke, and cardiovascular disease; for example, elevated levels of aMMP-8 in gingival grevicular fluid have been found to indicate poorly controlled diabetes patients.[3-6] Thus, the aMMP-8 PoC test can help medical professionals to screen their patients (with only 5-7 minutes) for elevated risk of periodontal diseases and refer them to dentist if the test is positive.
This has been amended to revised discussion, page 6, lines 193-196.
[1] Uitto VJ, Suomalainen K, Sorsa T. Salivary collagenase. Origin, characteristics and relationship to periodontal health. J Periodontal Res. 1990 May;25(3):135-42. https://doi.org/10.1111/j.1600-0765.1990.tb01035.x
[2] Gangbar S, Overall CM, McCulloch CA, Sodek J. Identification of polymorphonuclear leukocyte collagenase and gelatinase activities in mouthrinse samples: correlation with periodontal disease activity in adult and juvenile periodontitis. J Periodontal Res. 1990 Sep;25(5):257-67. https://doi.org/10.1111/j.1600-0765.1990.tb00914.x
[3] Lalla E, Papapanou PN. Diabetes mellitus and periodontitis: a tale of two common interrelated diseases. Nat Rev Endocrinol 2011;7:738-48. doi: 10.1038/nrendo.2011.106
[4] Safkan-Seppälä B, Sorsa T, Tervahartiala T, Beklen A, Konttinen YT. Collagenases in gingival crevicular fluid in type 1 diabetes mellitus. J Periodontol. 2006 Feb;77(2):189-94. https://doi.org/10.1902/jop.2006.040322
[5] Lafon A, Pereira B, Dufour T, Rigouby V, Giroud M, Béjot Y, Tubert-Jeannin S. Periodontal disease and stroke: a meta-analysis of cohort studies. Eur J Neurol 2014;21:1155-61, e66-7. doi: 10.1111/ene.12415
[6] Dietrich T, Sharma P, Walter C ym. The epidemiological evidence behind the association between periodontitis and incident atherosclerotic cardiovascular disease. J Clin Periodontol 2013;40 Suppl 14:S70-84. doi: 10.1111/jcpe.12062
Janket SJ, Baird AE, Chuang SK ym. Meta-analysis of periodontal disease and risk of coronary heart disease and stroke. Oral Surg Oral Med Oral Pathol Oral Radiol Endod 2003;95:559-69. https://doi.org/10.1067/moe.2003.107
Reviewer 2 Report
The authors compared the performance of aMMP-8 POC test to the conventional BOP method. The use of aMMP-8 POC can indeed improve patient experience and outcome. However, some important information are missing from this manuscript. For example, there is not sufficient information about how the aMMP-8 POC test really works. Presumably, the test is an immunoassay (lateral flow) method. In addition, how are the results interpreted? By naked eye or via an instrument? The manuscript will still benefit from further language editing. Here are some additional comments that should be addressed.
Line 48-49: The sentence needs to be reworded in order to improve clarity. Also, line 55-56 and 60-62.
Line 78: Please include a brief description of the method.
Table 1 and 2: There are more positive results for aMMP-8 test than the BOP test. If the BOP is the reference or gold standard method, then the higher positive results seen in the aMMP-8 test can be interpreted as false positive. This needs to be discussed.
Line 156-157: The sentence needs to be reworded in order to improve clarity.
Discussion section:
What are the limitations of using the aMMP-8 POC test?
Are there other alternative methods?
Author Response
Dear Reviewer 2,
Thank you for your comments!
Please find our responses to your comments below. They have been described in the revised manuscript, as well.
Yours faithfully,
Ismo Räisänen
Point 1: “However, some important information are missing from this manuscript. For example, there is not sufficient information about how the aMMP-8 POC test really works. Presumably, the test is an immunoassay (lateral flow) method. In addition, how are the results interpreted? By naked eye or via an instrument?”
Response 1: The test (PerioSafe®) is a point-of-care / chairside lateral flow collagenase-2 (aMMP-8) immunotest and is available as a visual and a quantitative test. This study used the visual test. Both tests (the visual and the quantitative) analyze patient’s mouthrinse and, thus, are a non-invasive diagnostic method causing no discomfort to patient. Mouthrinse was collected in three phases according to manufacturer’s instructions): first a 30 second prerinse with tap water, followed by a 60 second wait, and then rinsing with the test liquid (5 mL purified water) for 30 seconds and pouring it into a little collection cup. The test administrator collects 3 mL of the rinse into a syringe and places maximum of four drops in the lateral flow immunoassay system. The result can be read in 5-7 minutes as a color change resulting from immunoreactions: one single blue line for a negative result (no risk) and two blue lines for positive result (increased risk). Even a thin second line indicates increased risk for periodontitis and peri-implantitis. In the quantitative test, the test analyzes the concentration of aMMP-8 in patient’s mouthrinse. The visual and the quantitative tests have the same cut-off point for positives 20ng/mL. [1,2]
This has been amended to manuscript page 2 lines 77-82.
[1] Alassiri S, Parnanen P, Rathnayake N, Johannsen G, Heikkinen AM, Lazzara R, van der Schoor P, van der Schoor JG, Tervahartiala T, Gieselmann D, Sorsa T. The Ability of Quantitative, Specific, and Sensitive Point-of-Care/Chair-Side Oral Fluid Immunotests for aMMP-8 to Detect Periodontal and Peri-Implant Diseases. Dis Markers. 2018 Aug 5;2018:1306396. doi: 10.1155/2018/1306396
[2] Al-Majid A, Alassiri S, Rathnayake N, Tervahartiala T, Gieselmann DR, Sorsa T. Matrix Metalloproteinase-8 as an Inflammatory and Prevention Biomarker in Periodontal and Peri-Implant Diseases. Int J Dent. 2018 Sep 16;2018:7891323. doi: 10.1155/2018/7891323
Point 2: “Line 48-49: The sentence needs to be reworded in order to improve clarity. Also, line 55-56 and 60-62. Line 78: Please include a brief description of the method. Line 156-157: The sentence needs to be reworded in order to improve clarity.”
Response 2: Those lines have been edited to improve clarity and a brief description of the method has been amended. See page 2 lines 48-50, 54-58, 60-67 and lines 77-82 and page 5 line 153-155.
Point 3: “Table 1 and 2: There are more positive results for aMMP-8 test than the BOP test. If the BOP is the reference or gold standard method, then the higher positive results seen in the aMMP-8 test can be interpreted as false positive. This needs to be discussed.”
Response 3: There must be a misunderstanding here, because all the patients with positive test had signs of disease (≥4 mm periodontal pockets). Thus, the positive test results from the aMMP-8 PoC test cannot be false positives. In other words, it’s the other way around. BOP test gave more false negatives compared to the aMMP-8 PoC test. We have now included healthy control group (adolescents without any ≥4 mm periodontal pockets) (as requested by Reviewer 1) and updated our results section, tables and figures. We edited tables 1 and 2 and put together their information into a new table named “Table 1” to make the information more readable. See pages 3-5 lines 103-140.
Point 4: “Discussion section: What are the limitations of using the aMMP-8 POC test?
Are there other alternative methods?”
Response 4: Some limitations of using the aMMP-8 PoC test include patients with mixed dentition i.e. patients younger than 15 years old, active phase in orthodontic treatment, pericoronitis and a mouth ulcer. They are potential sources of error for the aMMP-8 PoC test in the identification of patients for periodontal diseases. In this study, adolescents had no such features. This has been amended to our manuscript, see page 5 lines 163-175.
Regarding alternative methods, they have been under development for other potential biomarkers in oral fluids for diagnostics of periodontitis and peri-implantitis. These biomarkers include, but are not limited to, neutrophil elastase, interleukin-1, myeloperoxidase, proteases of Porphyrmonas gingivalis and Treponema denticola (gingipains and dentilisin) and calprotectin. [1-3] It is possible to use them in combination, which may increase the accuracy of periodontitis diagnosis compared to a single biomarker. Unfortunately, these methods are still at experimental stage. On the other hand, visual and quantitative aMMP-8 PoC/ chairside assays, which function alone very well, are commercially available for dental and medical professionals linking these two disciplines. This has been amended to our manuscript, see page 6 lines 217-224.
[1] Ji S, Choi Y. Point-of-care diagnosis of periodontitis using saliva: technically feasible but still a challenge. Front Cell Infect Microbiol. 2015 Sep 3;5:65. doi: 10.3389/fcimb.2015.00065. eCollection 2015. Review.
[2] Srivastava N, Nayak PA, Rana S. Point of Care- A Novel Approach to Periodontal Diagnosis-A Review. J Clin Diagn Res. 2017 Aug;11(8):ZE01-ZE06. doi: 10.7860/JCDR/2017/26626.10411
[3] He W, You M, Wan W, Xu F, Li F, Li A. Point-of-Care Periodontitis Testing: Biomarkers, Current Technologies, and Perspectives. Trends Biotechnol. 2018 Nov;36(11):1127-1144. doi: 10.1016/j.tibtech.2018.05.013.
Round 2
Reviewer 1 Report
None
Reviewer 2 Report
no further comment